# Suicidal Ideation, Depression, Anxiety, Impulsivity, Self-Esteem, Emotional Regulation, Child Trauma and Hopelessness in Korean Military Soldiers

**DOI:** 10.3390/healthcare13182356

**Published:** 2025-09-18

**Authors:** Yeon Seo Lee, Youngil Lee, Myung Ho Lim

**Affiliations:** 1Department of Psychology, Graduate School, Dankook University, Cheonan 31116, Republic of Korea; dldustj0104@dankook.ac.kr; 2Department of Anatomy, College of Medicine, Dankook University, Cheonan 31116, Republic of Korea; anat104@dku.edu; 3Department of Psychology and Psychotherapy, College of Health Science, Dankook University, Cheonan 31116, Republic of Korea

**Keywords:** anxiety, childhood trauma, depression, emotional dysregulation, hopelessness, impulsivity, loneliness, military soldiers, self-esteem, stress, suicidal ideation

## Abstract

**Background/Objectives**: Suicide is the leading cause of death among South Korean military soldiers, accounting for more than 70% of all deaths. This issue is particularly relevant in the military context due to the nature of living in groups in a controlled environment. This study was conducted active-duty south Korean male soldiers aged 18 to 28 who were performing mandatory military service for one year and six months. Additionally, it compares and analyzes the differences in suicidal ideation and risk factors between military soldiers and a comparison group consisting of males in their 20s without military experience. **Methods**: This study included 248 Korean soldiers and 292 general controls, totaling 540 participants. The research instruments used for evaluation included the Beck Scale of Suicide Ideation (BSI), the Childhood Trauma Questionnaire (CTQ-SF), the Perceived Stress Scale (PSS), the Difficulties in Emotion Regulation Scale (DERS-16), the Barratt Impulsiveness Scale Version 11 (BIS-11), the Rosenberg Self-Esteem Scale (RSES), the Patient Health Questionnaire-9 (PHQ-9), the Generalized Anxiety Disorder 7-item scale (GAD-7), the UCLA Loneliness Scale (UCLAS), and the State-Beck Hopelessness Scale (S-BHS). **Results**: The results of this study showed that suicidal ideation, depression, anxiety, impulsivity, and self-esteem were significantly higher in the military group compared to the comparison group. Conversely, emotional dysregulation was considerably lower in the soldiers than in the comparison group. No significant differences were found in childhood trauma, stress, loneliness, and hopelessness between the two groups. Multiple regression analysis within the military group revealed that childhood trauma, hopelessness, and depression were major factors influencing suicidal ideation. **Conclusions**: These findings will help identify risk factors for suicide among soldiers and develop effective intervention strategies to prevent it.

## 1. Introduction

Suicidal ideation (SI) refers to the desire or having the intention to commit suicide [1]. Reducing SI is crucial for lowering the rates of suicide attempts and deaths [2]. Notably, suicide attempts are most prevalent among individuals in their 20s [1]. According to a study by Nock et al. [3] in most countries, SI first occurs in adolescence and young adulthood, increases rapidly, and then stabilizes in early middle age, with 33.6% of suicidal ideators planning suicide and 29.0% attempting suicide. In South Korea, males in their 20s are required to serve in the military if deemed eligible for active military service, with approximately 240,000 enlisting each year [4]. However, many soldiers cite challenges adjusting to military life, including perceived unfair treatment [5]. According to a 10-year statistical survey conducted by the Ministry of National Defense from 2014 to 2023, 610 soldiers died by suicide, accounting for 71.5% of all military deaths. This is still lower than the suicide rate for men in their 20s in Korea in 2022, which was 24.2 per 100,000 people, but the suicide rate among soldiers has been on the rise since 2022 [4]. Soldiers enlisting in their early 20s may lack fully developed coping mechanisms to handle pain and frustration, rendering them emotionally vulnerable to slight stressors that can jeopardize their mental health [6]. Fear stemming from unfamiliar environments, strict rules and discipline, a hierarchical organizational culture, work-related stress, interpersonal conflict, and isolation from their surroundings are some of the factors that make it difficult for soldiers to adjust to their units [6]. According to Ursano et al. [2] in the US, 44.2% of soldiers experiencing SI within the first month after deployment to a combat zone suffer from depression, while 19.3% endure stress-related challenges in military life. In addition, 6% of 85 soldiers with SI reported attempting suicide within 12 months of returning to their units. Suicide thoughts and depression of soldiers deployed in combat zones are so important that further research is needed. Previous studies on military suicide also included emotional depression as well as specific cognitive symptoms [2]. Suicide in the military can have significant repercussions within this close-knit organization [4]. Therefore, research on SI in the military is needed.

Cozolini [7] classified childhood physical and emotional abuse as complex trauma, as opposed to simple trauma, which is likely to be repeated. The researcher also reported that traumatic childhood experiences can manifest as various symptoms in adulthood, including suicidal behavior, even in response to mild triggers. Yoon et al. [8] reported that childhood trauma increased suicidal behavior in addition to other known risk factors like depression. Dube et al. [9] also found that psychiatric disorders partially mediated the relationship between childhood trauma and SI and suicidal behavior.

Shelef et al. [10] reported that subjective stress, especially in harsh situations, can further increase the risk of suicide and that psychological autopsies of suicide victims showed that suicide attempts often occurred shortly after stressful life events.

Goldner et al. [11] identified emotional dysregulation as a risk factor that could explain SI among soldiers.

Impulsivity is closely associated with controlling thoughts and behaviors and is correlated with the ability to conform to social norms [12,13]. In Table 1, the most common age group among soldiers is 20 years old, corresponding to late adolescence. This developmental stage is typically characterized by relative cognitive immaturity and limited emotional regulation, which may contribute to a heightened tendency toward impulsive behavior [6]. It has factors related to disciplinary problems and violent behavior in the military, and high impulsiveness can lead to behavioral problems such as suicide [14].

Self-esteem refers to the respect that one has for one’s own worth. Jeong et al. [15] reported that among soldiers classified as protective-concerned (and therefore at high risk for suicide), those with a history of SI and suicide attempts had higher levels of stress, depression, and hopelessness, along with lower self-esteem, compared to those without such experiences.

Depression is known to be the most powerful predictor of SI and has been shown to be strongly correlated with it [16,17]. Depression can lead to SI and other mental health problems, especially when individuals are exposed to high levels of stress reported that depression is linked to suicide, along with loss of motivation, helplessness, and cognitive decline [18,19]. In a study investigating suicide risk factors in the military, researchers at Florida State University found that anxiety was strongly correlated with suicide [18].

Loneliness is a painful subjective experience stemming from poor interpersonal relationships [20]. In a unique environment like the military, loneliness can trigger internal conflicts, leading to distress and maladjustment [18]. Beck et al. [21] defined hopelessness as a core characteristic of depression, characterized by a lack of enthusiasm and negative expectations for the future. Furthermore, Beck et al. [22] reported that hopelessness moderated the relationship between suicidal intent and depression in a study of suicide predictors, indicating it was 1.3 times more significant than depression in explaining SI. As an early warning signal, symptoms such as hopelessness can be an indicator for detecting suicide attempts. Based on this, through the development of intervention programs, it is possible to detect soldiers’ adjustment disorders and prevent suicide attempts [10]. Hopelessness is a critical measure for identifying groups at risk of suicide, with higher levels of hopelessness correlating to increased suicide risks [23].

In the present study, it is hypothesized that soldiers will exhibit higher levels of childhood trauma, stress, difficulties in emotion regulation, impulsivity, depression, anxiety, loneliness, hopelessness, and suicidal ideation compared to the comparison group, whereas self-esteem is expected to be lower than that of the comparison group.

Therefore, this study examined the differences in childhood trauma, stress, emotional regulation, impulsivity, self-esteem, depression, anxiety, loneliness, hopelessness, and SI among 248 Korean military soldiers and 292 adult males in their 20s without military experience to identify factors contributing to SI.

## 2. Materials and Methods

### 2.1. Participants and Procedures

Male soldiers stationed at a military base in City A participated in a direct survey as the military group. For the comparison group, a representative research institute, which has data on over 1.65 million individuals categorized by gender, region, and age in Korea, conducted an online survey targeting males in their 20s without military experience. The questionnaire included 164 items related to SI and other variables and took approximately 20 min to complete. The concept of SI was explained to the participants, and the survey was administered after obtaining their voluntary consent.

We excluded 28 people who showed insufficient or missing responses after data collection, and finally, data from a total of 540 participants (including 248 soldiers and 292 controls) were used for analysis. The military group was informed personally about the study and consented to participate, while the comparison group participated in the survey after voluntarily agreeing to take part through the survey institute. Participants who completed the survey and consented to the use of their personal information were given coupons worth approximately $1.4.

The demographic characteristics of the military and comparison groups are shown in Table 1.

### 2.2. Ethical Consideration and Informed Consent

This study was conducted after receiving approval from the Institutional Review Board of Dankook University (DKU 2025-03-033-002). Before the study began, the participants were fully informed about its purpose and contents. Voluntary consent was obtained after informing the participants that they could withdraw from the study at any time without any penalty if they experienced any psychological discomfort. They were also provided with a list of institutions to contact should they require psychological assistance.

### 2.3. Measures

In this study, several risk factors associated with SI were selected based on previous research, including childhood trauma, stress, emotional dysregulation, impulsivity, self-esteem, depression, anxiety, loneliness, and hopelessness.

#### 2.3.1. Beck Scale for Suicide Ideation (BSI)

The BSI, developed by Beck et al. [24] and validated in Korea by Lee and Kwon [25] was used to evaluate SI. This scale consists of 19 items measuring the presence and intensity of suicidal thoughts, such as plans, intensity, and duration, on a 3-point Likert scale, where higher scores indicate a greater tendency toward SI. The reliability was reported as 0.74 in Lee and Kwon [25] and 0.88 in this study.

#### 2.3.2. Childhood Trauma Questionnaire (CTQ-SF)

The CTQ-SF, developed by Bernstein et al. [26] and validated in Korea by Yu et al. [27] in 2009, consists of five subscales: emotional neglect (8 items), emotional abuse (5 items), physical neglect (5 items), physical abuse (5 items), and sexual abuse (5 items). In this study, 23 items were utilized from the CTQ-SF, excluding the subscale of sexual abuse (5 items). Each item was rated on a 5-point Likert scale. The reliability was 0.90 in Bernstein et al. [26] 0.79 in Yu et al. [27] and 0.92 in this study.

#### 2.3.3. Perceived Stress Scale (PSS)

The PSS, developed by Cohen et al. [28] in 1983, revised in 1988, and later adapted by Lee et al. [29] was used to measure stress levels. It consists of 10 items: 6 positive and 4 negative, which are rated on a 5-point Likert scale. The reliability was 0.82 in the study by Lee et al. [29] and 0.85 in this study.

#### 2.3.4. Difficulties in Emotion Regulation Scale (16-Item Version, DERS-16)

The DERS-16, a brief version of the DERS, was developed by Bjureberg et al. [30] based on the original 36-item DERS designed by Gratz and Roemer [31] in 2004. It consists of 16 items assessing various dimensions of emotional dysregulation: lack of emotional clarity (2 items), difficulties engaging in goal-directed behavior (3 items), impulse control difficulties (3 items), limited access to emotion regulation strategies (5 items), and nonacceptance of emotional responses (3 items). Respondents rate the extent to which each item applies to them on a 5-point Likert scale. The reliability was reported as 0.92 in Bjureberg et al. [30] and 0.96 in this study.

#### 2.3.5. Barratt Impulsiveness Scale-11 (BIS-11)

The Korean version of the BIS-11, developed by Barratt [10] and validated by Lee et al. [32] was used to measure impulsivity. This 30-item self-report instrument assesses three primary domains of impulsivity: attentional (10 items), motor (9 items), and non-planning (11 items). Each item is measured on a 4-point Likert scale. The reliability was 0.78 in Lee et al. [32] and 0.87 in this study.

#### 2.3.6. Rosenberg Self-Esteem Scale (RSES)

Self-esteem was measured using the RSES, developed by Rosenberg [33] in 1965 and adapted into Korean by Lee and Won [34]. It consists of 10 items, with five items each for positive and negative self-esteem, rated on a 4-point Likert scale. The reliability was 0.89 in the study by Lee and Won [34] and 0.80 in this study.

#### 2.3.7. Patient Health Questionnaire-9 (PHQ-9)

Depression was measured with the PHQ-9, which was developed by Spitzer et al. [35] in 1999 and adapted and validated by Park et al. [36] in 2010. Each item is rated on a 4-point Likert scale. The reliability at the time of development was 0.81, and 0.90 in this study.

#### 2.3.8. Generalized Anxiety Disorder 7-Item Scale (GAD-7)

The GAD-7 scale, developed by Spitzer et al. [37] in 2006, was used to measure the severity of anxiety. Each item is rated on a 4-point Likert scale. The reliability at the time of development was 0.92, and 0.92 in this study.

#### 2.3.9. UCLA Loneliness Scale (Version 3)

The UCLA Loneliness Scale (Version 3), developed by Russell et al. [38], later revised by Russell in 1996, and adapted and validated by Jin et al. [39] was used to measure loneliness. It consists of 20 items, comprising 9 positive and 11 negative items rated on a 4-point Likert scale. The reliability was 0.93 in the study by Jin et al. [39] and 0.93 in this study.

#### 2.3.10. State-Beck Hopelessness Scale (S-BHS)

The S-BHS, developed by Beck [21] and his research team and adapted and validated by Kim et al. [40] was used to measure hopelessness. It includes 20 questions that require a yes or no response, with higher scores indicating greater levels of hopelessness. The reliability was reported as 0.85 in Kim et al. [40] and 0.88 in this study.

### 2.4. Statistical Analysis

The collected data was analyzed using IBM SPSS Statistics 28.0. First, frequencies and percentages were calculated to assess the demographic characteristics. Second, Cronbach’s α values were computed to evaluate the reliability of each scale. Third, an independent samples MANCOVA was conducted to examine the differences between the military and comparison groups. And, multiple regression analysis was performed to investigate the relative influence of the variables.

To investigate differences in scores on childhood trauma, stress, emotion dysregulation, impulsivity, self-esteem, depression, anxiety, loneliness, hopelessness, and suicidal ideation between the military and non-military groups, a multivariate analysis of covariance (MANCOVA) was conducted, with income set as a covariate.

## 3. Results

### 3.1. A Multivariate Analysis of Covariance (MANCOVA)

The results of an MANCOVA to examine the mean differences in the study variables of childhood trauma, stress, emotion dysregulation, impulsivity, self-esteem, depression, anxiety, loneliness, hopelessness, and SI between the soldiers and the comparison group are summarized in Table 2. SI (F = 14.862 *, *p* = 0.000), depression (F = 10.391 *, *p* = 0.001), anxiety (F = 8.897 *, *p* = 0.003), impulsivity (F = 4.540 *, *p* = 0.034), and self-esteem (F = 4.654 *, *p* = 0.031) were significantly higher in the soldiers than in the comparison group. Conversely, emotion dysregulation (F = 7.589 *, *p* = 0.006) was significantly lower in the soldiers than in the comparison group. No significant differences were found between the two groups in terms of childhood trauma, stress, loneliness, and hopelessness.

### 3.2. Multiple Regression Analysis

Multiple regression analysis was conducted to examine the influence of the factors predicted to affect SI among soldiers: childhood trauma, stress, emotional dysregulation, impulsivity, self-esteem, depression, anxiety, loneliness, and hopelessness. The results indicated that childhood trauma (*t* = 5.888 *, *p* = 0.000), hopelessness (*t* = 5.704 *, *p* = 0.000), and depression (*t* = 2.743 *, *p* = 0.007) had a significant influence on SI. The detailed results for each variable are presented in Table 3.

## 4. Conclusions and Discussion

The main findings of this study are as follows.

First, SI, depression, anxiety, impulsivity, and self-esteem were significantly higher among soldiers than in the comparison group. This aligns with previous research indicating that mental health issues arising from maladaptation to a new environment are associated with SI [17,39,41]. According to reports from hospitalized soldiers, major depressive symptoms are a major factor in identifying soldiers at risk for SI [2,42]. This means that depression is strongly correlated with SI [16]. In addition, the stress of military life can lead to depression and hopelessness, and repeated exposure to negative circumstances may result in psychological issues such as suicidal impulses [43]. A meta-analysis of suicidal factors in military soldiers reported a significant relationship between depression and SI [6]. Given the strong connection between depression and SI, prioritizing treatment for depression is necessary for suicide prevention [16]. Other studies have reported that suicide attempts are most frequent within the first year of enlistment, accounting for 70% of all attempts. These attempts are often triggered by anxiety and conflicts stemming from sudden environmental changes, discomfort with a standardized lifestyle, and unfamiliar duties [44]. Fisher et al. [43] found that high anxiety levels are associated with increased vulnerability to stressful situations and a higher likelihood of suicide attempts. Soldiers are particularly susceptible to anxiety outside of the normal range during the first 2 to 4 weeks of basic military training after enlistment [45]. Soldiers face the challenges of adjusting to unit life, including strict rules and regulations, a hierarchical organizational culture, and the loneliness that comes from isolation from their surroundings. Especially in their early 20s, they may have a limited ability to process pain and frustration, making them emotionally vulnerable to even minor stressors [6]. Finally, impulsivity, defined as the tendency to act without thinking or planning, has been shown to be strongly associated with self-harm and suicidal behavior, along with depressive disorders [46]. Impulsivity is also a strong predictor of both SI and the transition from SI to suicide attempts [17]. This study also found strong correlations between SI and depression, anxiety, and impulsivity, indicating that mental health problems among soldiers may significantly influence SI.

However, the findings of this study did not align with previous research suggesting that self-esteem has a direct negative effect on SI and that lower self-esteem was associated with higher SI [47]. Instead, the results were consistent with a recent study by Joo et al. [48] which found that military service is increasingly perceived as an opportunity for career exploration and psychological growth, rather than merely a fulfillment of duty, and that higher pay and more opportunities for self-development as rank increases have a positive effect on self-esteem. Furthermore, emotional dysregulation was significantly lower in soldiers compared to the comparison group, which was inconsistent with a study by Goldner et al. [11] which reported that emotional dysregulation was associated with SI. However, another study noted that emotional dysregulation varies by rank and that psychological conflicts may trigger suicides, particularly during the initial adaptation phase [49]. This implies that an imbalance in emotional regulation over a certain period may influence SI. As such, SI manifests as a complex phenomenon influenced by various factors [19].

Among the risk factors for SI, childhood trauma, hopelessness, and depression were found to have significant effects on SI through regression analyses [16,22,50]. Research focused on military soldiers with suicidal impulses suggested that childhood bullying might indirectly influence SI [50]. In a study of SI among South Korean soldiers,. These findings were also consistent with other studies, which identify hopelessness and depression as major risk factors for suicide, and another study that highlight that experiencing SI can lead to unresolved psychological difficulties, an inability to escape hopelessness, and suicidal impulses [2,22,43]. In particular, soldiers have reported experiencing stress and despair when exposed to chaotic situations such as war and continuous violence, which increases the likelihood of SI and attempts [2,41].

Similarly, multiple regression analysis results for the comparison group showed that childhood trauma had a similar effect on mental health problems. In contrast, depression and hopelessness in the comparison group did not affect suicidal ideation.

This study has some limitations that suggest the need for follow-up research. First, participant were selected from a convenience sample of a specific military unit, the results may not be generalizable to the broader Korean military population. Therefore, future evaluations should consider these limitations. Although this study compared active-duty soldiers with adult males without military experience, it did not systematically consider socioeconomic factors such as rank and income. Future research should include these socioeconomic variables and systematically analyze Korean soldiers from all regions. Second, the questionnaires relied on self-report scales, which may have been influenced by participants’ unstable emotions, careless responses, or social desirability biases. Third, this study did not fully consider the unique characteristics of different environments. Future research should incorporate the distinct features of military organizational culture as well as the degree of soldiers’ adaptation to barracks life. Thus, further evaluation should take this into account.

Despite these limitations, this study is significant as it examines the relationship between SI and various identified risk factors among military soldiers, including childhood trauma, stress, emotional dysregulation, impulsivity, self-esteem, depression, anxiety, loneliness, and hopelessness. Furthermore, research on SI among military soldiers is not easily accessible due to the inherent challenges of collecting data within the military. Given the rarity of prior research on this topic, this study provides valuable insights through a comparative analysis of the risk factors of SI between soldiers and a general comparison group. In conclusion, this study identified childhood trauma, hopelessness, and depression as key risk factors for suicidal ideation among military soldiers and suggested the need for mental healthcare in the military setting, and the results of this study provide an empirical basis for the development of future suicide prevention programs for military soldiers. Future research should include military soldiers from different branches of the military and from all geographic regions to generate more generalizable findings. In order to prevent the suicide of soldiers, it is necessary to first identify risk factors, develop intervention programs, and apply these programs appropriately in the counseling field. In addition, these findings can be utilized in counseling and group therapy to help improve the emotional regulation of soldiers and improve suicidal impulses. For example, Linehan’s dialectical behavioral therapy (DBT) could help reduce soldiers’ suicide accidents.

## Figures and Tables

**Table 1 healthcare-13-02356-t001:** Demographic characteristics of the military and comparison groups.

Category		Soldiers		Comparison		T or X^2^ (*p*)
		N	(%)	N	(%)	
**Rank**	Private	38	(15.3)			
	Private first class	91	(36.7)			
	Corporal	89	(35.9)			
	Sergeant	30	(12.1)			
	Total	248	(100.0)			
**Age**	18	2	(0.8)	11	(3.8)	
	19	45	(18.1)	79	(27.1)	
	20	106	(42.7)	93	(31.8)	
	21	60	(24.2)	52	(17.8)	
	22	17	(6.9)	34	(11.6)	
	23	11	(4.4)	17	(5.8)	
	24	2	(0.8)	1	(0.3)	
	25	2	(0.8)	1	(0.3)	
	26	2	(0.8)	2	(0.7)	
	27	1	(0.5)	0	(0.0)	
	28	0	(0.0)	2	(0.7)	
	Total	248	(100.0)	292	(100.0)	
	M ± SD	20.37 ± 1.51		20.46 ± 1.319		−0.789 (0.430)
**Education**	High school graduate or lower	93	(37.5)	86	(29.5)	
	Attending college	143	(57.7)	192	(65.8)	
	College graduate or higher	12	(4.8)	14	(4.8)	
	Total	248	(100.0)	292	(100.0)	
						6.185 (0.186)
**Income**	KRW 1.18 M–2.82 M	39	(15.7)	77	(26.5)	
	KRW 2.82 M–4.36 M	60	(24.2)	42	(14.4)	
	KRW 4.36 M–6.36 M	86	(34.7)	71	(24.4)	
	KRW 6.36 M–11.54 M	44	(17.7)	72	(24.7)	
	KRW 11.54 M or more	19	(7.7)	29	(10.0)	
	Total	248	(100.0)	291	(100.0)	
						22.613 * (0.00)

* *p* < 0.05.

**Table 2 healthcare-13-02356-t002:** A multivariate analysis of covariance (MANCOVA) of the military and comparison groups.

Sortation	Soldiers (N = 248)	Comparison (N = 292)	*F*	*p*
	M ± SD	M ± SD		
**Childhood Trauma**	36.84 ± 16.771	36.61 ± 11.814	0.132	0.716
**Stress**	14.63 ± 8.782	14.92 ± 6.941	0.066	0.797
**Difficulties in Emotion Regulation**	30.33 ± 15.643	33.83 ± 12.667	7.589 *	0.006
**Impulsiveness**	61.96 ± 14.720	59.66 ± 10.997	4.540 *	0.034
**Self-Esteem**	29.38 ± 5.953	28.40 ± 4.504	4.654 *	0.031
**Depression**	6.22 ± 6.218	4.77 ± 4.687	10.391 *	0.001
**Anxiety**	4.48 ± 5.177	3.33 ± 4.062	8.897 *	0.003
**Loneliness**	18.16 ± 13.910	18.97 ± 11.284	0.582	0.446
**Hopelessness**	4.06 ± 4.740	4.01 ± 4.195	0.011	0.916
**Suicide Ideation**	4.73 ± 6.520	3.07 ± 3.440	14.862 *	0.000

* *p* < 0.05.

**Table 3 healthcare-13-02356-t003:** Multiple Regression Analysis of Military Soldiers’ Suicide Ideation.

Sortation	*B*	*S.E*	*β*	*t*	*p*	VIF
**(constant)**	−6.851	3.588		−1.910	0.057	
**Childhood Trauma**	0.128	0.022	0.330	5.888 *	0.000	1.915
**Stress**	−0.007	0.050	−0.010	−0.148	0.882	2.794
**Difficulties in Emotion Regulation**	0.009	0.031	0.021	0.285	0.776	3.339
**Impulsiveness**	−0.010	0.027	−0.023	−0.375	0.708	2.309
**Self-Esteem**	0.111	0.086	0.102	1.300	0.195	3.727
**Depression**	0.242	0.088	0.231	2.743 *	0.007	4.310
**Anxiety**	0.155	0.107	0.123	1.452	0.148	4.397
**Loneliness**	−0.005	0.041	−0.010	−0.111	0.912	4.687
**Hopelessness**	0.478	0.084	0.347	5.704 *	0.000	2.257
*R*^2^ = 0.609, Adjusted *R*^2^ = 0.595, *F*(*p*) = 41.240 (0.000)						

* *p* < 0.05, *B* = Unstandardized coefficient; *S.E* = Standard error; *β* = Standardized coefficient; VIF = Variance inflation factor.

## Data Availability

The original contributions presented in this study are included in the article. Further inquiries can be directed to the corresponding author(s). And the data is available only when there is a national or public request due to the Korean soldiers’ data.

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
