# Peer review of "Suicidal Ideation, Depression, Anxiety, Impulsivity, Self-Esteem, Emotional Regulation, Child Trauma and Hopelessness in Korean Military Soldiers"

_healthcare, 2025, doi:10.3390/healthcare13182356_

Round 1

Reviewer 1 Report

Comments and Suggestions for Authors

The manuscript under review is devoted to the important and practically significant problem on predictor of suicidal ideation in South Korean military soldiers. The purpose and content of the article correspond to scope of the journal Healthcare.

Overall, the research presented in the article makes a good impression in its design and execution. However, I believe that the authors need to correct a number of shortcomings in the presentation and discussion of the research results to improve the quality of the article.

I suggest that authors consider the following recommendations for improving the manuscript:

  • The Title of the article can be supplemented with: Features of Suicidal Ideation, Depression, Anxiety, Impulsivity Self-esteem, Emotional Regulation, Child Trauma and Hopelessness in South Korean Military Soldiers
  • The Abstract usually does not include the values of the criteria and their significance. At the same time, it is advisable to replace the last sentence (These findings will help identify risk factors for suicide among soldiers and develop effective intervention strategies to prevent it) with more specific recommendations.
  • A detailed review of relevant studies, starting with reference [40] in the reference list, should be moved to the Introduction. In the Discussion, these studies should be briefly mentioned in relation to the present research results.
  • In the description of this study, it is better not to use the terms "experimental" and "control" group, it is better to use, for example, "military group" and "comparison group".
  • Authors need to clarify in which group or groups the regression analysis was performed. 

The text states that the analysis was conducted "among soldiers" (lines 230-231), but Table 3 is titled “Multiple regression analysis of the experimental and control groups”. Judging by the description, the results of the regression analysis are presented only for the "soldier group". It is necessary to present the results of a similar regression analysis in the comparison group and further analyze the similarities and differences in significant predictors. Such a comparative analysis of significant predictors in the two groups is necessary for the completeness of the study's conclusions.

  • In Conclusion, the authors would like to formulate detailed recommendations on how the research findings can be used to prevent suicide among Korean soldiers.

Author Response

Response to Reviewer 1

Q1- Reviewer Comment

The Title of the article can be supplemented with: Features of Suicidal Ideation, Depression, Anxiety, Impulsivity Self-esteem, Emotional Regulation, Child Trauma and Hopelessness in South Korean Military Soldiers

Q1- Response

Thank you so much. I have taken your valuable feedback into account and have incorporated the relevant content accordingly.

[Before]

Suicidal Ideation, Depression, Anxiety, Impulsivity Self-esteem, Emotional Regulation, Child Trauma and Hopelessness in Korean Military Soldiers

[After]

Features of Suicidal Ideation, Depression, Anxiety, Impulsivity, Self-esteem, Emotional Regulation, Child Trauma and Hopelessness in South Korean Military Soldiers

Q2- Reviewer Comment

The Abstract usually does not include the values of the criteria and their significance. At the same time, it is advisable to replace the last sentence (These findings will help identify risk factors for suicide among soldiers and develop effective intervention strategies to prevent it) with more specific recommendations.

Q2- Response

Thank you. we based on your kind feedback, I intend to organize as follows.

The results of this study showed that suicidal ideation (t = -3.618*, p = .000), depression (t = -3.017*, p = .003), anxiety (t = -2.856*, p = .004), impulsivity (t = -2.029*, p = .043), and self-esteem (t = -2.114*, p = .035) were significantly higher in the military group compared to the comparison group. Conversely, emotional dysregulation (t = 2.821*, p = .005) was considerably lower in the soldiers than in the comparison group. No significant differences were found in childhood trauma, stress, loneliness, and hopelessness between the two groups. Multiple regression analysis within the military group revealed that childhood trauma (t = 5.888*, p = .000), hopelessness (t = 5.704*, p = .000), and depression (t = 2.743*, p = .007) were major factors influencing suicidal ideation. [line 27]

Thank you. for your valuable feedback. In response, we would like to remove the following statement to the manuscript.

These findings will help identify risk factors for suicide among soldiers and develop effective intervention strategies to prevent it [line 35]

Q3- Reviewer Comment

A detailed review of relevant studies, starting with reference [40] in the reference list, should be moved to the Introduction. In the Discussion, these studies should be briefly mentioned in relation to the present research results.

Q3- Response

Thank you. I based on your valuable feedback, I intend to organize as follows.

[Before]

Reports from soldiers in inpatient care demonstrate a high prevalence of depressive symptoms among those who have experienced SI, corroborating findings that major depressive symptoms are critical in identifying the risk of SI following combat deployment [268 í–‰]

[After]

According to reports from hospitalized soldiers, major depressive symptoms are a major factor in identifying soldiers at risk for SI. This means that depression is strongly correlated with SI. [line 272]

Yoon et al[8] reported that childhood trauma increased suicidal behavior in addition to other known risk factors like depression. Dube et al[9] also found that psychiatric dis-orders partially mediated the relationship between childhood trauma and SI and sui-cidal behavior. [ line312→line 72]

Q4- Reviewer Comment

In the description of this study, it is better not to use the terms "experimental" and "control" group, it is better to use, for example, "military group" and "comparison group".

.Q4- Response

Thank you for your valuable feedback. I've made all the necessary revisions. You can check them in the research content.

Q5- Reviewer Comment

Authors need to clarify in which group or groups the regression analysis was performed.

The text states that the analysis was conducted "among soldiers" (lines 230-231), but Table 3 is titled “Multiple regression analysis of the experimental and control groups”. Judging by the description, the results of the regression analysis are presented only for the "soldier group". It is necessary to present the results of a similar regression analysis in the comparison group and further analyze the similarities and differences in significant predictors. Such a comparative analysis of significant predictors in the two groups is necessary for the completeness of the study's conclusions.

Q5- Response

In the revised manuscript, the multiple regression table for the comparison group was not inserted. (In general, we collected statistical experts' opinions that it would be better to describe only the regression table with only the experimental group-military soldiers group.) However, the comparison results with the military group were described descriptively in the discussion.

Similarly, multiple regression analysis results for the comparison group showed that childhood trauma had a similar effect on mental health problems. In contrast, depression and hopelessness in the comparison group did not affect suicidal ideation.[line 322]

Q6- Reviewer Comment

In Conclusion, the authors would like to formulate detailed recommendations on how the research findings can be used to prevent suicide among Korean soldiers.

Q6- Response

Thank you. for your valuable feedback. In response, we would like to respond as follows.

In order to prevent the suicide of soldiers, it is necessary to first identify risk factors, develop intervention programs, and apply these programs appropriately in the counseling field. In addition, these findings can be utilized in counseling and group therapy to help improve the emotional regulation of soldiers and improve suicidal impulses. For example, Linehan's dialectical behavioral therapy (DBT) could help reduce soldiers' suicide accidents. Furthermore, This study can be used as a basis for policy formulation to promote mental health and prevent suicide among military soldiers. [line 354]

Reviewer 2 Report

Comments and Suggestions for Authors

Thank you for the opportunity to review the manuscript entitled “Suicidal Ideation, Depression, Anxiety, Impulsivity Self-esteem, Emotional Regulation, Child Trauma and Hopelessness in Korean Military Soldiers” (healthcare-3784007).

The paper explores the associations between suicidal ideation and some risk factors among active-duty Korean soldiers. The research included 248 Korean soldiers and 292 general controls.

The topic is highly relevant and timely, given the mental health challenges faced by military populations. The inclusion of both military and general population samples adds value to the work. However, several aspects of the manuscript require substantial revision before it can be considered for publication.

  • Title: The current title is overly long and dispersive due to the extensive list of variables. I recommend rephrasing it in a more concise and focused way.
  • Abstract: It is generally advisable to avoid including statistical indices (e.g., t, p) in the abstract, to improve readability.
  • Abstract: The abstract should also provide more detailed information about the participants, (e.g., gender distribution, age range), in order to help readers contextualize the sample used in the study.
  • Keywords: Keywords should be listed in alphabetical order, following standard practice.
  • Introduction: This section requires substantial expansion. The risk factors are only briefly introduced; each construct (e.g., depression, hopelessness, impulsivity) should be more thoroughly discussed in light of existing literature to support the rationale for their inclusion.
  • Introduction: Based on the literature, the authors should consider formulating clear hypotheses to guide the study.
  • Method: More detailed information is needed about the recruitment procedure, setting, and administration of the questionnaires.
  • Method: The authors should explicitly state the significance threshold (e.g., p < .05), and indicate whether any correction for multiple comparisons (e.g., Bonferroni correction) was applied.
  • Discussion and Conclusions: The practical implications of the findings should be more thoroughly addressed. I suggest creating a dedicated section to emphasize their relevance for clinical and military mental health practice.

Best regards,

Author Response

Response to Reviewer 2

Q1- Reviewer Comment

Title: The current title is overly long and dispersive due to the extensive list of variables. I recommend rephrasing it in a more concise and focused way.

Q1- Response

Thank you so much. We based on your valuable feedback, I intend to organize this contents.

To express it concisely, it could be expressed as “mental health among Koreans,” but the title is broad and abstract, so I thought it would be better to use a precise variable name, and I revised it with reference to the opinion of “reviewer 1”.

Q2- Reviewer Comment

Abstract: It is generally advisable to avoid including statistical indices (e.g., t, p) in the abstract, to improve readability.

Q2- Response

Thank you. I based on your valuable feedback, I intend to organize as follows.

The results of this study showed that suicidal ideation (t = -3.618*, p = .000), depression (t = -3.017*, p = .003), anxiety (t = -2.856*, p = .004), impulsivity (t = -2.029*, p = .043), and self-esteem (t = -2.114*, p = .035) were significantly higher in the military group compared to the comparison group. Conversely, emotional dysregulation (t = 2.821*, p = .005) was considerably lower in the soldiers than in the comparison group. No significant differences were found in childhood trauma, stress, loneliness, and hopelessness between the two groups. Multiple regression analysis within the mil-itary group revealed that childhood trauma (t = 5.888*, p = .000), hopelessness (t = 5.704*, p = .000), and depression (t = 2.743*, p = .007) were major factors influencing suicidal ideation. [line 27]

Q3- Reviewer Comment

Abstract: The abstract should also provide more detailed information about the participants, (e.g., gender distribution, age range), in order to help readers contextualize the sample used in the study.

Q3- Author's Response

Thank you, we based on your valuable feedback, I intend to organize as follows.

In response to the reviewer’s valuable feedback, we have now included information regarding participants’ gender and age in the revised manuscript.

before:

This study aims to determine suicidal ideation among active-duty Korean soldiers and explore the major risk factors.[line 13]

after:

This study was conducted on active-duty South Korean male soldiers aged 18 to 28 who were performing mandatory military service for one year and six months.[line 17]

Q4- Reviewer Comment

Keywords: Keywords should be listed in alphabetical order, following standard practice.

Q4- Response

Yes, based on your valuable feedback, I intend to organize the following content.

The keywords were arranged in alphabetical order in accordance with standard practice, as recommended in the feedback.

before:

Keywords: Suicidal ideation; childhood trauma; stress; emotional dysregulation; im-pulsivity; self-esteem; depression; anxiety; loneliness; hopelessness; military soldiers [line 37]

after:

Keywords: anxiety; childhood trauma; depression; emotional dysregulation; hopeless-ness; impulsivity; loneliness; military soldiers; self-esteem; stress; Suicidal ideation [line 37]

Q5- Reviewer Comment

Introduction: This section requires substantial expansion. The risk factors are only briefly introduced; each construct (e.g., depression, hopelessness, impulsivity) should be more thoroughly discussed in light of existing literature to support the rationale for their inclusion.

Q5- Author's Response

Yes, based on your valuable feedback, I intend to organize as follows.

Based on existing literature, explanations for each risk factor were added to provide supporting evidence.

Suicide thoughts and depression of soldiers deployed in combat zones are so important that further research is needed. Previous studies on military suicide also included emotional depression as well as specific cognitive symptoms.[2] [line 64 ]

As an early warning signal, symptoms such as hopelessness can be an indicator for detecting suicide attempts. Based on this, through the development of intervention programs, it is possible to detect soldiers' adjustment disorders and prevent suicide attempts.[8] [line 109 ]

It has factors related to disciplinary problems and violent behavior in the military, and high impulsiveness can lead to behavioral problems such as suicide.[12]. [line 89

Q6- Reviewer Comment

Introduction: Based on the literature, the authors should consider formulating clear hypotheses to guide the study.

Q6- Response

Thank you, I based on your valuable feedback, I intend to organize as follows.

As suggested in the feedback, the study hypotheses were incorporated as follows

In the present study, it is hypothesized that soldiers will exhibit higher levels of childhood trauma, stress, difficulties in emotion regulation, impulsivity, depression, anxiety, loneliness, hopelessness, and suicidal ideation compared to the comparison group, whereas self-esteem is expected to be lower than that of the comparison group.[line 115]

Q7- Reviewer Comment

Method: More detailed information is needed about the recruitment procedure, setting, and administration of the questionnaires.

Q7- Response

Yes, based on your valuable feedback, I intend to organize a follows.

The recruitment procedures, as well as the setup and administration, were described as follows.

Male soldiers stationed at a military base in A City participated in a direct survey as the military group. For the comparison group, a representative research institute, which has data on over 1.65 million individuals categorized by gender, region, and age in Korea, conducted an online survey targeting males in their 20s without military ex-perience. The questionnaire included 164 items related to SI and other variables and took approximately 20 minutes to complete. The concept of SI was explained to the participants, and the survey was administered after obtaining their voluntary consent.

We excluded 28 people who showed insufficient or missing responses after data collection, and finally, data from a total of 540 participants (including 248 soldiers and 292 controls) were used for analysis.

The military group was informed personally about the study and consented to participate, while the comparison group participated in the survey after voluntarily agreeing to take part through the survey institute. Participants who completed the survey and consented to the use of their personal information were given coupons worth approximately $ 1,4. [line 127]

Q8- Reviewer Comment

Method: The authors should explicitly state the significance threshold (e.g., p < .05), and indicate whether any correction for multiple comparisons (e.g., Bonferroni correction) was applied.

Q8- Author's Response

Thank you, I based on your valuable feedback, I intend to organize as follows.

In our data, type 1 errors may appear due to multiple comparisons. However, Bonferroni correction was not performed. However, the table and data describe that MANCOVA corrected for the covariate was used.

before:

The results of the independent samples t-test to examine the differences in the means of the study variables of childhood trauma, stress, emotional dysregulation, impulsivity, self-esteem, depression, anxiety, loneliness, hopelessness, and SI between soldiers and the comparison group are summarized in Table 2.

SI (t = -3.618*, p = .000), depression (t = -3.017*, p = .003), anxiety (t = -2.856*, p = .004), impulsivity (t = -2.029*, p = .043), and self-esteem (t = -2.114*, p = .035) were significantly higher in the soldiers compared to the comparison group. In contrast, emotional dysregulation (t = 2,821*, p = .005) was considerably lower in the soldiers than in the comparison group. No significant differences were found in childhood trauma, stress, loneliness, and hopelessness be-tween the two groups.[line 242]

after:

To investigate differences in scores on childhood trauma, stress, emotion dysreg-ulation, impulsivity, self-esteem, depression, anxiety, loneliness, hopelessness, and sui-cidal ideation between the military and non-military groups, a multivariate analysis of covariance (MANCOVA) was conducted, with income set as a covariate[line 234]

3.1. A multivariate analysis of covariance (MANCOVA) between the soldiers and the comparison group [ line 239]

Table 2. A multivariate analysis of covariance (MANCOVA) between the soldiers and the comparison group [line 253]

The results of an independent-samples F-test mancova to examine the mean dif-ferences in the study variables of childhood trauma, stress, emotion dysregulation, impulsivity, self-esteem, depression, anxiety, loneliness, hopelessness, and SI between the soldiers and the comparison group are summarized in Table 2. SI (F = 14.862*, p = .000), depression (F = 10.391*, p = .001), anxiety (F = 8.897*, p = .003), impulsivity (F = 4.540*, p = .034), and self-esteem (F = 4.654*, p = .031) were significantly higher in the soldiers than in the comparison group. Conversely, emotion dysregulation (F = 7.589*, p = .006) was significantly lower in the soldiers than in the comparison group. No sig-nificant differences were found between the two groups in terms of childhood trauma, stress, loneliness, and hopelessness.[line 242]

Q9- Reviewer Comment

Discussion and Conclusions: The practical implications of the findings should be more thoroughly addressed. I suggest creating a dedicated section to emphasize their relevance for clinical and military mental health practice.

Q9- Response

Yes, based on your valuable feedback, I intend to organize as follows.

In order to prevent the suicide of soldiers, it is necessary to first identify risk factors, develop intervention programs, and apply these programs appropriately in the counseling field. In addition, these findings can be utilized in counseling and group therapy to help improve the emotional regulation of soldiers and improve suicidal impulses. For example, Linehan's dialectical behavioral therapy (DBT) could help reduce soldiers' suicide accidents.

Reviewer 3 Report

Comments and Suggestions for Authors

1-)please be sure that the information is correct:

Suicide is the leading cause of death among South Korean mili- 11
tary soldiers, accounting for more than 70% of all deaths

2-)be sure that criticizing South Korean military is allowed in South Korea.

3-)also , be sure that you have relevant permission to conduct the study.

4-)please mention the duration of military duty in South Korea.

5-)please give more information about the reference:

. Although this figure is lower 47
compared to the general population, the suicide rate among military soldiers has been 48
steadily increasing since 2022.[4] 

6-)be sure the following is citeable and scientific. if not please remove it:

Military soldiers are often in late 71
adolescence, a period characterized by cognitive immaturity and limited emotional regu- 72
lation, which makes them prone to exhibit impulsive behavior

7-)please be sure that the tables are consistent.

8-)the discussion section seems very limited also its unclear why soldiers are under pressure in the context of your study.

9-)please add suggestions to improve it and reduce depression and suicide rate.

10-)be sure that following sentence is necessary.

 This study 317
can be used as a basis for policy formulation to promote mental health and prevent suicide 318
among military soldiers. 

11-)add limitations related to your study.

Author Response

Response to Reviewer 3

Q1- Reviewer Comment

Comments and Suggestions for Authors

1-)please be sure that the information is correct:

Suicide is the leading cause of death among South Korean military soldiers, accounting for more than 70% of all deaths

Q1- Response

Thank you for your valuable review. This is based on statistical data investigated by the Ministry of National Defense. The relevant information is included in the introduction.

According to a 10-year statistical survey conducted by the Ministry of National Defense from 2014 to 2023, 610 soldiers died by suicide, accounting for 71.5% of all military deaths. [line 49]

Q2- Reviewer Comment

2-)be sure that criticizing South Korean military is allowed in South Korea.

Q2- Response

Thank you for your valuable review.

This study was conducted with the approval of the Bioethics Committee (IRB). In addition, negative descriptions such as suicide-related research results were also reviewed by military internal representatives.

Q3- Reviewer Comment

3-) also , be sure that you have relevant permission to conduct the study.

Q3- Response

Thank you for your valuable review.

Since the research was conducted with the permission of the military and approval from the bioethics committee (IRB) of the affiliated school, there was no problem in conducting the research.

Q4- Reviewer Comment

4-)please mention the duration of military duty in South Korea.

Q4- Author's Response

Thank you for your valuable review. As you said, the content has been revised as follows based on feedback.

before:

This study aims to determine suicidal ideation among active-duty Korean soldiers and explore the major risk factors.[line 15]

after:

The purpose of this study was to identify suicidal ideation and explore key risk factors among active-duty Korean male soldiers aged 18 to 28,who typyically serve a mandatory military service of 1 year and 6 monthes.[line 15]

after:

This study was conducted active-duty south Korean male soldiers aged 18 to 28 who were performing mandatory military service for one year and six minths [ line 17]

Q5- Reviewer Comment

5-) please give more information about the reference:

Although this figure is lower compared to the general population, the suicide rate among military soldiers has been steadily increasing since 2022.[4]

Q5- Author's Response

Yes, based on your valuable feedback, I intend to organize as follows content.

This is still lower than the suicide rate for men in their 20s in Korea in 2022, which was 24.2 per 100,000 people, but the suicide rate among soldiers has been on the rise since 2022.[line 51]

Q6- Reviewer Comment

6-)be sure the following is citeable and scientific. if not please remove it:

Military soldiers are often in late adolescence, a period characterized by cognitive immaturity and limited emotional regu-lation, which makes them prone to exhibit impulsive behavior

Q6- Author's Response

Yes, based on your valuable feedback, I intend to organize the following content.

before:

Military soldiers are often in late adolescence, a period characterized by cognitive immaturity and limited emotional regulation, which makes them prone to exhibit im-pulsive behavior. Notably, impulsivity and aggressive traits can be significant predic-tors of suicide

after:

In Table 1, the most prevalent age group among soldiers was 20 years, which corresponds to late adolescence. This developmental stage is typically characterized by relative cognitive immaturity and limited emotional regulation, which may contribute to a heightened tendency toward impulsive behavior.[6][line 82]

Q7- Reviewer Comment

7-)please be sure that the tables are consistent.

Q7- Response

Thank you for your valuable review.

In accordance with the reviewer's delicate opinions, the result values of the table were neatly organized and simplified to increase the readability of the table.

Q8- Reviewer Comment

8-)the discussion section seems very limited also its unclear why soldiers are under pressure in the context of your study.

Q8- Author's Response

Thank you for your valuable review.

This content was previously described in the introduction section, and the following has been added to the discussion section.

[introduction]

Soldiers enlisting in their early 20s may lack fully developed coping mechanisms to handle pain and frustration, rendering them emotionally vulnerable to slight stressors that can jeopardize their mental health.[6] Fear stemming from unfamiliar environ-ments, strict rules and discipline, a hierarchical organizational culture, work-related stress, interpersonal conflict, and isolation from their surroundings are some of the factors that make it difficult for soldiers to adjust to their units.[6] [line 55]

[discussion]

Soldiers face the challenges of adjusting to unit life, including strict rules and regulations, a hierarchical organizational culture, and the loneliness that comes from isolation from their surroundings. Especially in their early 20s, they may have a limited ability to process pain and frustration, making them emotionally vulnerable to even minor stressors.[6] [line 288]

Q9- Reviewer Comment

9-)please add suggestions to improve it and reduce depression and suicide rate.

Q9- Author's Response

Thank you for your valuable review.

As reviewer 2 mentioned I added the following to the conclusion

In order to prevent the suicide of soldiers, it is necessary to first identify risk factors, develop intervention programs, and apply these programs appropriately in the counseling field. In addition, these findings can be utilized in counseling and group therapy to help improve the emotional regulation of soldiers and improve suicidal impulses. For example, Linehan's dialectical behavioral therapy (DBT) could help reduce soldiers' suicide accidents.

Moreover, these research results can be utilized in individual counseling and group therapy to improve emotional control, reduce suicidal thoughts, and alleviate relationship conflicts by introducing Linehan's dialectical behavior therapy (DBT) to enhance the mental health of soldiers.[line 339]

Q10- Reviewer Comment

10-)be sure that following sentence is necessary.

This study can be used as a basis for policy formulation to promote mental health and prevent suicide among military soldiers.

Q10- Author's Response

We sincerely appreciate your action. We will delete the information [line 361].

after: We sincerely appreciate the reviewers' valuable comments. While we understand the suggestion to remove this information, please consider the author's opinion that the results of this paper can serve as a useful reference for developing suicide prevention policies.

Q11- Reviewer Comment

11-)add limitations related to your study.

Q11- Response

Thank you for your review. In the conclusion, a third limitation was additionally addressed.

Third, this study did not fully consider the unique characteristics of different environments. Future research should incorporate the distinct features of military organizational culture as well as the degree of soldiers’ adaptation to barracks life. [line 317]

This study has limitations, which suggest the need for further research. Because participants were selected from a convenience sample of a specific military unit, the results may not be generalizable to the broader Korean military population. Therefore, future evaluations should consider these limitations.[line 327]

Round 2

Reviewer 1 Report

Comments and Suggestions for Authors

I thank the authors for their attention to my comments. I see that almost all my recommendations were taken into account and adjustments were made to the text.

I do not fully understand why the table with the results of the regression analysis in the comparasion group was not included in the text. However, I leave this issue to the discretion of the authors and the Academic Editor.

I also recommend that the authors carefully edit the text.

Author Response

We apologize again for not including the regression analysis table for the comparison group in the main text.
Instead, we will include it in detail and describe it in an appendix to our dissertation paper.
Thank you again for your thoughtful feedback.

Reviewer 3 Report

Comments and Suggestions for Authors

accept

Author Response

Thank you again for your kind and thoughtful feedback.